# Sowing Depth Monitoring System for High-Speed Precision Planters Based on Multi-Sensor Data Fusion

**DOI:** 10.3390/s24196331

**Published:** 2024-09-30

**Authors:** Song Wang, Shujuan Yi, Bin Zhao, Yifei Li, Shuaifei Li, Guixiang Tao, Xin Mao, Wensheng Sun

**Affiliations:** 1College of Engineering, Heilongjiang Bayi Agricultural University, Daqing 163319, China; wshbau@163.com (S.W.); zhaobin@byau.edu.cn (B.Z.); lyifei@126.com (Y.L.); lsflsf2000@163.com (S.L.); tgx1996@163.com (G.T.); mx1631@163.com (X.M.); 17382791810@163.com (W.S.); 2Provincial Key Laboratory of Intelligent Agricultural Machinery Equipment, Daqing 163319, China; 3College of Engineering, Northeast Agricultural University, Harbin 150030, China

**Keywords:** high-speed no-till seeder, sowing depth monitoring, improved sparrow search algorithm, extended Kalman filter, data fusion

## Abstract

High-speed precision planters are subject to high-speed (12~16 km/h) operation due to terrain undulation caused by mechanical vibration and sensor measurement errors caused by the sowing depth monitoring system’s accuracy reduction problems. Thus, this study investigates multi-sensor data fusion technology based on the sowing depth monitoring systems of high-speed precision planters. Firstly, a sowing depth monitoring model comprising laser, ultrasonic, and angle sensors as the multi-sensor monitoring unit is established. Secondly, these three single sensors are filtered using the Kalman filter. Finally, a multi-sensor data fusion algorithm for optimising four key parameters in the extended Kalman filter (EKF) using an improved sparrow search algorithm (ISSA) is proposed. Subsequently, the filtered data from the three single sensors are integrated to address the issues of mechanical vibration interference and sensor measurement errors. In order to ascertain the superiority of the ISSA-EKF, the ISSA-EKF and SSA-EKF are simulated, and their values are compared with the original monitoring value of the sensor and the filtered sowing depth value. The simulation test demonstrates that the ISSA-EKF-based sowing depth monitoring algorithm for high-speed precision planters, with a mean absolute error (*MAE*) of 0.083 cm, root mean square error (*RMSE*) of 0.103 cm, and correlation coefficient (*R*) of 0.979 achieves high-precision monitoring. This is evidenced by a significant improvement in accuracy when compared with the original monitoring value of the sensor, the filtered value, and the SSA-EKF. The results of a field test demonstrate that the ISSA-EKF-based sowing depth monitoring system for high-speed precision planters enhances the precision and reliability of the monitoring system when compared with the three single-sensor monitoring values. The average *MAE* and *RMSE* are reduced by 0.071 cm and 0.075 cm, respectively, while the average *R* is improved by 0.036. This study offers a theoretical foundation for the advancement of sowing depth monitoring systems for high-speed precision planters.

## 1. Introduction

The term sowing depth is used to describe the vertical distance at which a seed or a plant seedling is placed in soil. This is a crucial factor influencing the growth and development of crops. An appropriate sowing depth can provide seeds with the necessary moisture, temperature, and oxygen, as well as other essential growth conditions, thus promoting seed germination, plant development, and root growth, in addition to further enhancing the growth potential and yield of crops. The in-depth research on sowing depth monitoring technology, as well as the achievement of precise sowing depth control or water replenishment, has significant theoretical and practical implications for the promotion of precision agriculture and the improvement of crop yield [1,2,3,4]. During the high-speed operations of no-till planters, a number of factors may affect the accuracy and reliability of the sowing depth monitoring system. These include mechanical vibration caused by terrain fluctuations, sensor measurement errors, and the lack of reliability of single-sensor monitoring systems. Therefore, the development of an accurate and reliable sowing depth monitoring system is of particular importance. A number of scholars have employed sensor technology in sowing depth monitoring. However, the influence of numerous interfering factors persists in impeding the advancement of this field, particularly in terms of enhancing the precision and dependability of sowing depth monitoring. Research aiming to optimise sensor performance, minimise environmental interference, and refine data processing algorithms will prove instrumental in advancing sowing depth monitoring systems, and multi-sensor data fusion [5] represents an effective approach to strengthening such systems.

The sowing depth monitoring methods are mainly divided into two categories. The first category comprises monitoring methods that use mechanical-type sensors [6]. Through the experimental calibration and derivation of the establishment of a planter to an underground pressure model, the changes in the lower pressure are measured and used to deduce the changes in sowing depth so as to monitor the sowing depth [7]. Li et al. [8] studied a no-tillage planter sowing depth intelligent adjustment system based on flex sensors and the Mamdani fuzzy model. It used three flex sensors to monitor the downward pressure exerted by the planter on the ground, and the peak output voltage increased linearly with the increasing downward force, allowing for the monitoring of the sowing depth. Gao et al. [9,10,11] investigated a CAN bus monitoring and evaluation system for precision planters’ downforce and sowing depth. Angle and axle pin sensors were used as the sowing depth measuring devices, and the sowing depth could be reflected by measuring the force on the ground and the position of the furrow opener. Optimised designs of the hydraulic and pneumatic drive devices with partition control allowed for the monitoring control of the operating parameters and a quality evaluation based on CAN bus communication. Static modelling tests of the sowing depth and downforce were completed using an indoor test bench, and a downforce measurement model that can adapt to different sowing depth settings was established.

The other category comprises monitoring methods that use distance sensors. Regardless of whether sowing depth is measured directly or indirectly, these methods do not require complex theoretical mechanical models. Furthermore, during the operation of the sowing machine, the use of the seed box seed and fertiliser box will be reduced, which may have a certain impact on mechanical models. Additionally, during high-speed operations of the monobloc, the bumps encountered will cause the bouncing phenomenon, and thus, the reliability of such sowing depth monitoring systems poses a challenge. Galibjon et al. [12] proposed a total station measurement of sowing depth using different current excitation magnetorheological damper coils, different sowing depths, and standard error analyses to verify the sowing depth dynamic measurement accuracy of sowing monitoring devices. Wen et al. [13] proposed an automatic control system for the sowing depth of precision planters, and it uses ultrasonic sensors to collect real-time position signals to output the target depth in order to accurately monitor sowing depth. Nielsen et al. [14] developed a monitoring system that can capture changes in sowing depth and adjust it in real-time to improve sowing efficiency, using angle sensors to monitor changes in the angle of the furrow opener in order to determine the changes in the sowing depth. Pasi et al. [15] proposed a depth control system for planters based on the ISO11783 standard [16], installing a system to measure mechanical joint angles and ultrasonic sensors to conduct direct distance measurements in order to monitor sowing depth changes.

Aiming to address the problems of sensor measurement error [17,18,19], reduced precision of the sowing depth monitoring system caused by a large amount of noise interfering with the sensor measurement accuracy [20] in the field, and the poor reliability of single-sensor monitoring [21], this study investigates a high-speed no-tillage planter sowing depth monitoring system based on the ISSA-EKF. In this system, signals from laser, ultrasonic, and angular sensors are collected separately and filtered, and then the filtered signal data are fused to improve their precision and reliability.

## 2. Materials and Methods

### 2.1. Sowing Depth Monitoring Methods

In a no-tillage seeder in field operation, the sowing unit carries out the breaking of stubble, furrowing, sowing, and mulching compaction, comprising four links [21]. Among them, the furrowing link is the process by which sowing depth is determined. Under the action of the seeder’s own gravity and the depth-limiting mechanism, the ground surface is placed at the bottom of the sowing unit disc furrow opener, and the theoretical depth of the fixed seed furrow is determined. A schematic of the sowing depth formation is shown in Figure 1.

However, the field terrain is uneven, and high-speed operation under the conditions of single-body bouncing will lead to an inconsistent sowing depth. In this case, active compensation is needed, and a sowing depth monitoring value is needed for the active compensation system to feed back the sowing depth; thus, the monitoring of the reliability of the sowing depth directly affects the control accuracy of the active compensation system. The sowing machine selected for this study is the 1205-type high-speed no-tillage seeder produced by Beijing Debang Dawei Company. The disc opener of this planter and the sowing unit are relatively fixed at the same sowing depth position; however, due to the ground reaction force on the depth-limiting wheel, there is a height difference with the disc opener. Therefore, the actual sowing depth refers to the height difference between the bottom of the depth-limiting wheel and the bottom of the disc opener. The four links of the sowing unit swing on the axis of the fixed beam at the front end of the frame, and the upper and lower links always move in parallel. During operation, the four-rod profiling mechanism ensures that the sowing unit maintains up and down flat movements when working in the field so as to ensure the consistency of the sowing depth. In order to accurately measure the sowing unit sinking depth variable, a laser sensor, an ultrasonic sensor, and an angle sensor are selected to conduct simultaneous measurements. A schematic of the sowing depth monitoring sensor arrangement is shown in Figure 2.

The three sensors of the research broadcast depth monitoring sensor group are the GJD-03 laser sensor (Ruixingjia Electronic, Shenzhen, China). in Guangdong Province, with a working voltage of 5~24 V DC; the UB1000-18GM75 ultrasonic sensor(Jiamei Science and Technology, Wenzhou, China) with a working voltage of 9~15 V DC; and the DYL626S angle sensor(Yongheng Science and Technology, Wuxi, China) with a working voltage of 10~30 V DC. The above sensors are Modbus sensors based on RS485 communication. The sowing depth monitoring system circuit is shown in Figure 3. The sensors’ main parameters are shown in Table 1. Diagrams of the detection principles of the sensors in the sowing depth detection system are shown in Figure 4.

A laser sensor is installed between the disc opener and the suppression wheel, and the laser emission point of the sensor is installed parallel to the lowest point of the disc opener in order to ensure that it can measure the deepest part of the sowing furrow. When the sowing monobloc with the disc opener removed is placed in the soil, the distance between the laser emission probe and the compaction surface, as measured by the laser sensor, is 18.42 cm. Assuming that, during the no-tillage planter in-soil operation, the laser sensor measurement is *H_L_*, the formula for the sowing depth *H* is
(1)$$H=H_{L}-18.42$$

A sensor-fixing platform is installed at the front end of the frame of the sowing unit, and the ultrasonic sensor probe installation position of this platform is parallel to the ground. Additionally, an ultrasonic reflective plate, which is horizontal to the ground, is installed at the side of the sowing unit, and the reflective plate is 20.43 cm away from the ultrasonic transmitting end of the sensor when the sowing unit with the disc opener removed is placed in the soil. Assuming that, during the operation of the no-tillage planter in the soil, the ultrasonic sensor measurement is *H_U_*, the formula for the sowing depth *H* is
(2)$$H=H_{U}-20.43$$

With the parallel installation of an angle sensor on the four parallel links of the sowing unit, the centre distance between the two connection points of the four links is 40.0 cm. When the sowing unit with the unloaded disc opener is placed in the soil, there is a slight height difference between the four-link mechanism and the actual horizontal plane, and its actual height should be subtracted from the height difference. It can be seen that there is a difference of 1.55 cm, as measured by the sensor. Additionally, it is assumed that, in the case of no-tillage sowing machine operation in the soil, the operation swing angle is *α*; therefore, the formula of the sowing depth *H* is
(3)H=40sinα−1.55

In general, the noise interference contained in the acquisition signal of the laser sensors is due to the reflection and scattering of the laser beam [22]. The noise interference contained in the acquisition signal of the ultrasonic sensors is due to the ambient noise and acoustic interference when the ultrasonic sensor receives the acoustic wave for echo measurement [23]. The noise interference contained in the acquisition signal of the angle sensor is due to electromagnetic interference, including that from mechanical vibration, power lines, and high-frequency devices [24]. Thus, the measurement accuracy is reduced. In conclusion, the laser, ultrasonic, and angle sensors are susceptible to mechanical interference, environmental noise, and measurement errors during the high-speed operation of the planter, which leads to a reduction in monitoring accuracy. In order to solve the above problems, multi-sensor fusion algorithms are widely used in multi-sensor monitoring systems with strong interference to reduce the effects of errors and noise on the sensor measurement results and improve measurement accuracy. The extended Kalman filter (EKF), which has the advantages of good real-time performance, robustness, high accuracy, and the ability to handle a large amount of data, has been widely used in sensor monitoring and other fields [25,26].

### 2.2. Sowing Depth Raw Data Collection

Sowing depth raw data were collected from 25 May to 29 May 2024 in the test field on the southeast side of Heilongjiang Bayi Agricultural University, Daqing City, Heilongjiang Province (125°166′~125°168′ E, 46°581′~46°584′ N). The test field was a no-tillage plot with unturned topsoil. In a pre-test plot, soil compaction was determined at 360 points equally spaced 2 m apart at 5 cm depths using a JC-JSD-01 soil compactness meter produced by Qingdao Juchuang Environmental Protection Group, Qingdao, China, and the average compactness was 7.1 kg/cm^2^. The test was conducted using a John Deere 484 tractor (John Deere, Moline, IL, USA) towing a Deppon Dawei 1205 no-tillage planter (Deppon Dawei, Beijing, China) with an average operating speed of 12–16 km/h. The sowing depth data of 360 points were collected manually as the theoretical sowing depth values, as well as the original data to be filtered and fused with the sowing depth values in the simulation test. Due to the limitation of the length of the plot, it was necessary to repeat the test 10 times from the starting point of each operation, measuring the sowing depth value at an equal spacing of 2 m. Additionally, Hall sensors were arranged on the ground wheel of the planter so that when the ground wheel started to rotate, the three single sensors turned on to conduct synchronous measurements, and every time it travelled 2 m, the three single sensors conducted synchronous measurements once. A Tianmu XG-150 high-precision digital display depth gauge (Tianmu, Guilin, China) was used to measure the number of collection points output by the system after each test. The sowing depth values of the first and last five points in the 10 tests were from the acceleration and deceleration phases of the tractor, so the middle points of the 10 tests, totalling 360 effective collection points, were used as the original data for the theoretical measurement value in the simulation test, and the sowing depth data obtained from the three single sensors in each test were used for the same operation. To compare the actual monitoring effect of the three single sensors, this study selected three key performance assessment indices: the mean absolute error (*MAE*), the root mean square error (*RMSE*), and the correlation coefficient (*R*). The *MAE* and *RMSE* were used to assess the accuracy of the sensor monitoring values, and the *R* was used to assess the reliability of the sensor monitoring values in order to verify the validity of the multi-sensor method of measuring sowing depth. A comparison of the performance evaluation indicators between the measured values and the sensors’ real values is shown in Table 2, and comparison graphs of the three single sensors’ measured values and the manually measured values are shown in Figure 5.

In Table 1 and Figure 5, it can be seen that the three sensors perform relatively well in terms of the *MAE*, *RMSE*, and *R* key performance indicators. This means that the differences between the three measurements and the real values are small, with high consistency and stability. Therefore, the measurements of the three sensors can be considered reliable and suitable for subsequent filtering and data fusion operations.

### 2.3. EKF Multi-Sensor Fusion Algorithm

The Kalman filter reduces the effects of noise and errors on data through a feedback mechanism; however, its performance is affected by the linear nature of the system and noise. A system or measurement model with a nonlinear component may not be able to handle the problem with a Kalman filter. However, the EKF, a Kalman filter algorithm based on nonlinear systems [27], can transform the nonlinear component of the system into a linear component through linearisation and updating the observation and state transition equations. In data fusion, the Kalman filter results from multiple sensors can be fused to estimate the system state using their corresponding metric models to obtain a more accurate and reliable sowing depth.

For complex scenarios such as sowing operations, efficient filtering and algorithm fusion are required to eliminate noise interference and improve the estimation accuracy and tracking capability of the target state. The signals collected from the laser, ultrasonic, and angle sensors are processed using Kalman filtering, and then the filtered data are fused using the EKF to obtain more accurate and reliable sowing depth values. The EKF has higher accuracy when applied to a sowing depth measurement system in this environment; it can handle the problem of a large amount of data and has scalability. Furthermore, for the processing of complex sensor data, the EKF can improve the sensor tracking capabilities.

The state and measurement equations are mathematical equations used to describe the EKF dynamic system and the observation process. The state quantity is the sowing depth estimate, and the observation quantity is the sowing depth measurement obtained using multiple sensors. The state quantity *x* is defined as the sowing depth estimate, and the state transfer function is defined according to the physical model and system dynamics. For the state transfer equation, the sowing depth can be considered a first-order lagging system with the state equation of
(4)$$x_{k}=x_{k-1}+w_{k}\;,\left(w_{k}\sim N\left(0,Q\right)\right)$$
where *x_k_* is the sowing depth state vector at moment *k*; *x_k_*_−1_ is the sowing depth state vector at moment *k* − 1; and *w_k_* is the system process noise at time *k*, which obeys Gaussian noise with a mean of 0 and covariance *Q.*

The EKF measurement equation is obtained from the previous sowing depth measurement equation for the three single sensors’ measurements:(5)$$z_{k}=\left[\begin{array}{c} H_{u}\\ a\\ H_{L} \end{array}\right]=\left[\begin{array}{c} x_{k}+18.42\\ \arcsin(\frac{x_{k}+1.55}{40})\\ x_{k}+20.43 \end{array}\right]+v_{k},\left(v_{k}\sim N\left(0,R\right)\right)$$
where *z_k_* is the observation vector at moment *k*, and *v_k_* is the measurement noise at moment *k.*

This, in turn, leads to the EKF-based sowing depth monitoring equation:(6)$${\hat{x}}_{k|k-1}={\hat{x}}_{k-1|k-1}$$
(7)$${\hat{P}}_{k|k-1}={\hat{P}}_{k-1|k-1}+Q_{k}$$
where ${\hat{x}}_{k|k-1}$ is the priori estimate of the sowing depth at moment *k*, cm; ${\hat{x}}_{k-1|k-1}$ is the *k* − 1 posteriori estimate of the sowing depth at moment *k* − 1, cm; ${\hat{P}}_{k|k-1}$ is the priori covariance matrix at moment *k*; ${\hat{P}}_{k-1|k-1}$ is the vector *k* − 1 moment a posteriori covariance matrix; and *Q_k_* is the monitoring state Gaussian noise *k* moment covariance matrix.

As the monitoring model of the angle sensor is nonlinear, it must be linearised; thus, the EKF linearisation, measurement, and monitoring equations are
(8)$$H_{k}=\left[\begin{array}{c} 1\\ \frac{1}{\sqrt{40^{2}-{\left({\hat{x}}_{k|k-1}+1.55\right)}^{2}}}\\ 1 \end{array}\right]$$
(9)$${\hat{z}}_{k|k-1}=\left[\begin{array}{c} {\hat{x}}_{k|k-1}+18.42\\ \arcsin(\frac{{\hat{x}}_{k|k-1}+1.55}{40})\\ {\hat{x}}_{k|k-1}+20.43 \end{array}\right]\;$$
where *H_k_* is the measurement matrix, and ${\hat{z}}_{k|k-1}$ is the sensor observation variables at moment *k*.

The EKF update allows the sowing depth data monitored during the fusion process to be fused with the system model to obtain a more accurate estimate of the state, and the updated EKF equation is
(10)$$K={\hat{P}}_{k-1|k-1}H_{k}^{T}{\left(H_{k}{\hat{P}}_{k-1|k-1}H_{k}^{T}+R_{k}\right)}^{-1}$$
(11)$${\hat{x}}_{k|k}={\hat{x}}_{k|k-1}+K\left(z_{k}-{\hat{z}}_{k|k-1}\right)$$
(12)$${\hat{P}}_{k|k}=\left(1-KH_{k}\right){\hat{P}}_{k|k-1}$$
where *K* is the Kalman gain; $H_{k}^{T}$ is the transpose matrix of the measurement matrix; and *R_k_* is the Gaussian noise *k* moment covariance array of the measured values.

The EKF can estimate the sowing depth using the above operations, and the trade-offs in determining the *Q* and *R* values need to be adjusted according to specific application scenarios and system characteristics. In general, the system noise covariance matrix (*Q*) reflects the uncertainty in the model of the sowing depth measurement system, and the measurement noise covariance matrix (*R*) reflects the measurement uncertainty of the three single sensors. When *Q* is set too large, it leads to an overdependence of the filter on the model and, thus, insensitivity to changes in the measurements, which may result in a slower filter response. Conversely, when *Q* is set too small, the filter becomes overly dependent on the measurements and may be disturbed by measurement noise, leading to unstable results. Therefore, when selecting *Q*, a balance between the model and the measurement is required, and when selecting *R* and *Q*, they need to be rationally adjusted according to the actual measurement error.

### 2.4. Improved Sparrow Search Algorithm

The *R* matrix and *Q* matrix parameter rectification of the EKF is actually a parameter optimisation problem, and the use of the intelligent swarm optimisation algorithm is an efficient way to solve it. The SSA is a new type of swarm intelligent optimisation algorithm; it searches the solution space of the optimisation problem through a simulation of the sparrow population behavioural strategy, and it simulates the life behaviour of the sparrow population to search for the optimum [28]. Compared with other algorithms, in optimising the EKF problem, the SSA increases the diversity of the search by moving and communicating between individuals, thus having the ability to perform a global search and avoid falling into a local optimum. Secondly, the SSA has parallelism and distributed computation abilities, which can search multiple candidate solutions in *R* and *Q* matrices at the same time. The SSA also has adaptability and robustness, and it can be better adapted to different optimisation problems. In addition, the SSA is scalable and applicable to various optimisation scenarios, such as parametric, functional, and combinatorial optimisation. The principle of multi-sensor data fusion in the sowing depth monitoring system is shown in Figure 6.

In the SSA, the explorer is responsible for exploring new potential solutions and ensuring the global search capability of the algorithm. The position update equation of the explorers is
(13)$$X_{i}^{t+1}=X_{i}^{t}\cdot \exp(-\beta \cdot t)\cdot \exp(\frac{R_{2}}{ST}\cdot \xi )$$
where *X_i_* is the position of the *i*-th sparrow, *t* is the current number of iterations, *β* is the control parameter, *R*_2_ is a random number between [0, 1], *ST* is the safety threshold, and *ξ* is the Gaussian distributed random number.

The followers in the SSA enhance the local search by following the optimal solution to improve the convergence speed of the algorithm, and the position update equation of the followers is
(14)$$X_{i}^{t+1}=\frac{X_{i}^{t}+\sum_{j=1}^{n}\left(X_{j}^{i}-X_{i}^{t}\right)}{n}$$
where *n* is the total number of sparrows.

The scouts in the SSA, however, participate in the search process while maintaining the security of the group, which increases the diversity and robustness of the algorithm. The position update equation of the scouts is
(15)$$X_{i}^{t+1}=X_{i}^{t}+\xi \cdot X_{best}^{t_{i}^{t}}$$
where *X*_best_ is the location of the current optimal solution.

Although the SSA has many technical advantages in the parameter rectification problem of the *R* and *Q* matrices of EKF filters, it also has some disadvantages, such as initialisation parameter setting and cross-selection assignment problems. Therefore, this study improves the SSA by introducing the chaotic mapping algorithm [29] to adjust the initialisation process of the SSA. The aim of the initialisation formula is to determine the initial position of an individual in each dimension by mapping the result of the chaotic mapping to the position of a sparrow. Chaotic mapping is extremely sensitive to the initial conditions, and even small differences in the initial values can lead to completely different iteration sequences. This property allows chaotic mapping to produce diverse search paths in optimisation algorithms, helping to prevent the algorithm from falling into local optima. Using ICMIC chaotic mapping,
(16)$$z_{n+1}=z_{n}\cdot \sin(z_{n})$$
where *z_n_* is the chaos variable.

Meanwhile, the Gaussian random walk algorithm [30] is used to optimise cross-selection allocation. Gaussian random walk can strike a balance between exploring new regions and exploiting known information by controlling the size of the standard deviation. A small standard deviation facilitates local search, while a large standard deviation promotes global search. Gaussian random walk allows for fine-grained exploration in the vicinity of the current solution, which helps the SSA to search deeper in the vicinity of the local optimum, thus increasing the probability of finding a better solution. The randomness provided by Gaussian random walk helps the algorithm to resist the effects of noise and outliers, and it improves the robustness of the algorithm. The Gaussian random walk formula introduced in the crossover process is given as
(17)$$X_{i}^{t+1}=X_{i}^{t}\cdot \exp(-\beta \cdot t)\cdot \exp(\frac{R_{2}}{ST}\cdot \xi +\sigma \cdot \varepsilon )$$
where *σ* is the standard deviation of the Gaussian random walk, and *ε* is the Gaussian distributed random number.
(18)$$X_{i}^{t+1}=\frac{X_{i}^{t}+\sum_{j=1}^{n}\left(X_{j}^{i}-X_{i}^{t}\right)+\sigma \cdot \varepsilon }{n}$$
(19)$$X_{i}^{t+1}=X_{i}^{t}+\xi \cdot X_{best}^{t_{i}^{t}\cdot }$$

In this way, chaotic mapping and Gaussian random walk are integrated into the SSA to improve its search performance and ability to solve complex optimisation problems.

### 2.5. ISSA-EKF

This study selected the *RMSE* as the fitness function for the optimisation of the algorithm; the selection of the *RMSE* as the fitness function can help the ISSA to find a more desirable solution in the optimisation of the EKF problem in order to improve the accuracy and reliability of the model for sowing depth monitoring.
(20)$$fitness=\sqrt{\frac{1}{n}\overset{n}{\underset{i=1}{\sum }}\left(s_{i}-w_{i}\right)}$$
where *fitness* is the fitness function; *s_i_* is the actual sowing depth, cm; *w_i_* is the estimated sowing depth, cm; and *n* is the sampling time.

According to the previous analysis, the ISSA optimises the parameters of the *Q* and *R* matrices in the EKF, where the covariance matrix *Q* of process noise is a one-dimensional matrix, and the covariance matrix *R* of measurement noise is a three-dimensional matrix; thus, it optimises a total of four parameters, namely, *Q_σ_* in the *Q* matrix and *R_σ_*_1_, *R_σ_*_2_, and *R_σ_*_3_ in the *R* matrix. The ISSA-EKF has five operational steps: the initialisation of the optimisation environment, crossing behaviour, population update, termination conditions, and output results. Through these steps, the optimisation of the parameters of the *Q* and *R* matrices in the EKF by the ISSA can be achieved to improve the accuracy and reliability of the model monitoring. A flowchart of the ISSA-EKF is shown in Figure 7.

## 3. Results

### 3.1. Simulation Test

The simulation and comparison test environment of the ISSA-EKF-based sowing depth monitoring system for high-speed no-tillage planters consisted of an Intel(R) Core(TM) i5-9300H CPU@2.40GHz, 8G RAM, 64-bit Windows 11 operating system, and Matlab2020b. The simulation test compared the Kalman-filtered sowing depth obtained from the sowing depth values collected from the three single sensors with the sowing depth obtained after the fusion of two intelligent optimisation algorithms. Based on previous experience, the Kalman filter process noise variance *Q_k_* was set to 0.05, and the observation noise variance *R_k_* was set to 0.01; the above five sowing depths were still compared in terms of the three key performance evaluation indices of the *MAE*, *RMSE*, and *R* to verify the superiority of the ISSA-EKF. Because there were four optimisation parameters in the simulation test, in order to avoid a too-long computing time, the parameter optimisation range was narrowed down to shorten the optimisation time, and the parameter optimisation range was set for the four key parameters based on the previous tuning experience of the EKF before the start of the simulation comparison test, where *Q_σ_* ∈ [0, 10,000], *R_σ_*_1_ ∈ [0, 10,000], *R_σ_*_2_ ∈ [0, 10,000], and *R_σ_*_3_ ∈ [0, 10,000]; the population size of the SSA and ISSA was 50, and the maximum number of iterations was 100 generations. A comparison of the results of the five key indicators of monitoring algorithms is shown in Table 3. A comparison of five filtering algorithms and manual measurements is shown in Figure 8. A diagram comparing the absolute error of eight monitoring methods is shown in Figure 9.

As can be seen in Table 3 and Figure 8 and Figure 9, the Kalman filtering of the three single sensors resulted in a slight increase in the accuracy and reliability of the monitoring results; however, there was no extremely significant improvement compared with the three single sensors’ monitoring results: the average *MAE* was reduced by 0.006 cm, the average *RMSE* was reduced by 0.017 cm, and the average *R* was improved by 0.004. The two intelligent fusion algorithms’ results were greatly improved compared with the three single sensors’ sowing depth monitoring results: the average *MAE* was reduced by 0.067 cm, the average *RMSE* was reduced by 0.070 cm, and the average *R* was improved by 0.036. The two intelligent fusion algorithms’ results showed a greater improvement than the three single sensors’ sowing depth monitoring results after filtering: the average *MAE* was reduced by 0.061 cm, the average *RMSE* was reduced by 0.050 cm, and the average *R* was improved by 0.032. Regarding the sowing depth data based on the SSA-EKF, the *MAE* was 0.096 cm, the *RMSE* was 0.120 cm, and the *R* was 0.971. Regarding the sowing depth data based on the ISSA-EKF, the *MAE* was 0.083 cm, the *RMSE* was 0.103 cm, and the *R* was 0.979; that is, the ISSA-EKF reduced the *MAE* by 0.013 cm, reduced the *RMSE* by 0.017 cm, and improved the *R* by 0.008 compared with the SSA-EKF. Therefore, the proposed ISSA-EKF is able to accurately monitor the sowing depth during the high-speed sowing operations of no-tillage seeders. Furthermore, compared with the monitoring results of the three single sensors, the filtered sowing depth with the SSA-EKF has a highly significant improvement in both accuracy and reliability. To compare the performance of the ISSA with that of the SSA, the optimisation ability of the two optimisation algorithms was compared, and the results are shown in Table 4. Variation curves of the fitness values of the two algorithms are shown in Figure 10. Iterative curves of the key parameters of the SSA for optimising the EKF are shown in Figure 11. Iterative curves of the key parameters of the ISSA for optimising the EKF are shown in Figure 12.

In Table 4 and Figure 10, Figure 11 and Figure 12, it can be seen that the SSA-based algorithm optimises the EKF with a fitness value of 0.134 at the initial optimisation and reaches the optimal fitness value at the 20th generation with a value of 0.120; the ISSA-based algorithm optimises the EKF with a fitness value of 0.130 at the initial optimisation and reaches the optimal fitness value at the 90th generation with a value of 0.103. The four parameter cases of the SSA-optimised EKF are as follows: *Q_σ_* reaches the optimum in the 20th generation, with an optimum value of 395; *R_σ_*_1_ reaches the optimum in the 20th generation, with an optimum value of 68; *R_σ_*_2_ reaches the optimum in the 20th generation, with an optimum value of 60; and *R_σ_*_3_ reaches the optimum in the 11th generation, with an optimum value of 0.03. The parameter case of the ISSA-EKF is as follows: *Q_σ_* reaches the optimum in the 90th generation, with an optimum value of 6400 and a fitness value of 0.103. The performance of the ISSA is much better than that of the SSA in the optimisation of the initial population, which indicates that the initial solution generated by the SSA with chaotic mapping is better than the initial solution generated by the SSA alone. Regarding the optimisation ability, the ISSA outperforms the SSA in terms of the degree of optimisation and optimal adaptation. This indicates that the Gaussian random walk algorithm outperforms the SSA’s own mutation behaviour in terms of the cross-mutation link.

### 3.2. Field Test

A field validation test of the ISSA-EKF-based sowing depth monitoring system for high-speed no-till planters was conducted from 15 to 17 June 2024 in the experimental field of Heilongjiang Bayi Agricultural University, Daqing City, Heilongjiang Province (125°166′~125°168′ E, 46°581′~46°584′ N). Before the test, the soil firmness was determined in pre-test plots using a JC-JSD-01 soil firmness meter at 360 points equally spaced 2 m apart at depths of 3 cm, and the average firmness was 3.9 kg/cm^2^. The soil firmness was also determined in the pre-test plots at 360 points equally spaced 2 m apart at depths of 5 cm, and the average firmness was 6.8 kg/cm^2^. The test was conducted using a John Deere 484 tractor (John Deere, Moline, IL, USA) towing a Deppon Dawei 1205 no-tillage planter (Deppon Dawei, Beijing, China) with an average operating speed of 12~16 km/h. The test was repeated 10 times from the start of each operation, measuring the sowing depth value at a spacing of 2 m. A Tianmu XG-150 high-precision digital display depth gauge (Tianmu, Guilin, China) was used to measure the number of collection points output by the system after each test. The sowing depth values from the first and last five points of the 10 tests were from the acceleration and deceleration phases of the tractor; thus, they were removed, and the middle points of the 10 tests, totalling 360 effective collection points, were used. Additionally, the sowing data obtained from the three single sensors in each test were used for the same processing. In order to verify the system versatility of the algorithm at different operating depths, tests were conducted to verify the monitoring accuracy of the monitoring system at block sowing depths of 5 cm and 3 cm, and the two groups collected 360 points. The field test is shown in Figure 13, and the results of the field test are shown in Table 5.

In Table 5, it can be seen that, under different theoretical sowing depth operating conditions, the average *MAE* of the ISSA-EKF is reduced by 0.071 cm compared with that of all three single sensors; the average *RMSE* of the ISSA-EKF is reduced by 0.075 cm compared with that of all three single sensors; and the average *R* of the ISSA-EKF is improved by 0.036 compared with that of all three single sensors. Even when the data from the three single sensors are averaged, the error metrics are still higher than those of the ISSA-EKF. This again demonstrates that the ISSA-EKF has a higher sowing depth monitoring accuracy, achieved by fusing information from multiple sensors.

### 3.3. Discussion

The use of single sensors causes large errors in sowing depth monitoring. The large data errors and low reliability of the three single sensors indicate that the monitoring accuracy of a single sensor is limited. The ISSA-EKF, however, significantly improves the sowing depth monitoring accuracy by mining and fusing information from multiple sensors. The *MAE* and *RMSE* of the ISSA-EKF are significantly lower than those of the single sensors, whereas the *R* is higher, indicating that it has a significantly better monitoring effect than the single sensors. It is also difficult to compensate for the individual errors of a single sensor using averaging, and the error index of the ISSA-EKF is still much lower than the average error of the three single sensors, indicating that the averaging of data obtained from a single sensor cannot improve its overall monitoring effect.

Compared with a single sensor, the ISSA-EKF can give fuller play to the value of multi-sensor information and significantly improve the precision and reliability of sowing depth monitoring, as it can identify and mine the hidden data patterns of a single sensor and efficiently fuse multi-source sowing depth information. Therefore, the ISSA-EKF is a high-precision sowing depth monitoring method that can improve the accuracy and reliability of sowing depth monitoring systems. According to the NY/T1143-2006 Technical Specification for Quality Evaluation of Sowing Machines [31], when the sowing depth is greater than 3 cm, the sowing depth is qualified within ±1 cm, and this study can significantly reduce the magnitude of the measurement error, which can improve the control accuracy of the active profiling device.

This study concerns the selection of RS485 bus communication sensors, which can be used as a communication interface with multiple sensors. In the future, for the expansion of the multi-row sowing monomer sowing depth monitoring system, it may be beneficial to consider interface access to the same monomer arrangement of sensors. This would allow for the achievement of different sub-systems of individual communication, thus preventing the different sowing monomer information flows from interfering with each other. Furthermore, prospective enhancements to the hardware system’s reliability and the algorithmic structure will be considered. The objective is to achieve this without compromising monitoring accuracy. One potential avenue for improvement is the removal of the laser and ultrasonic sensors, which are direct measurement sensors that require a fixed frame. This would reduce the system’s overall space requirements while simultaneously lowering the hardware costs and system complexity.

## 4. Conclusions

(1) A multi-sensor sowing depth monitoring system is constructed with a laser sensor, an ultrasonic sensor, and an angle sensor, and a mathematical model for the sowing depth monitoring of this monitoring unit is established. The three single sensors are filtered separately using the Kalman filter. A multi-sensor data fusion algorithm is proposed to optimise the key parameters (*Q_σ_*, *R_σ_*_1_, *R_σ_*_2_, and *R_σ_*_3_) in the EKF, and this is achieved by incorporating the chaotic mapping algorithm and the Gaussian random walk algorithm of the ISSA of the multi-sensor data fusion algorithm to fuse the filtered data of the three sensors. Ultimately, this solves the problems of the reduced precision of sowing depth monitoring systems due to mechanical vibration and sensor measurement errors caused by the undulation of the terrain during the high-speed operation of the no-till planter, as well as the poor reliability of single-sensor monitoring.

(2) A simulation test shows that the ISSA-EKF-based sowing depth monitoring algorithm for high-speed no-tillage planters has an *MAE* of 0.083 cm, an *RMSE* of 0.103 cm, and an *R* of 0.979, and it achieves a high degree of accuracy compared with the manually measured data. The sowing depth accuracy after data fusion is significantly improved compared with that of both the original sensor monitoring value and the filtered monitoring value. The ISSA outperforms the SSA in the initial population searching ability, while the ISSA outperforms the SSA in the optimisation ability. The results also verify the optimisation effect of adding chaotic mapping and the Gaussian random walk algorithm to the SSA.

(3) The results of a field test show that the ISSA-EKF-based sowing depth monitoring algorithm for high-speed no-till planters reduces the average *MAE* and the average *RMSE* by 0.071 cm and 0.075 cm, respectively, compared with the monitoring values of the three single sensors, while the average *R* improves by 0.036. Compared with the single sensors, the ISSA-EKF can give fuller play to the value of multiple-sensor information, significantly improving the accuracy and reliability of sowing depth monitoring. This is due to the fact that the ISSA-EKF identifies and mines the hidden data patterns of the single sensors and efficiently fuses multi-source information. Therefore, the ISSA-EKF is a high-precision sowing depth monitoring method.

## Figures and Tables

**Figure 1 sensors-24-06331-f001:**
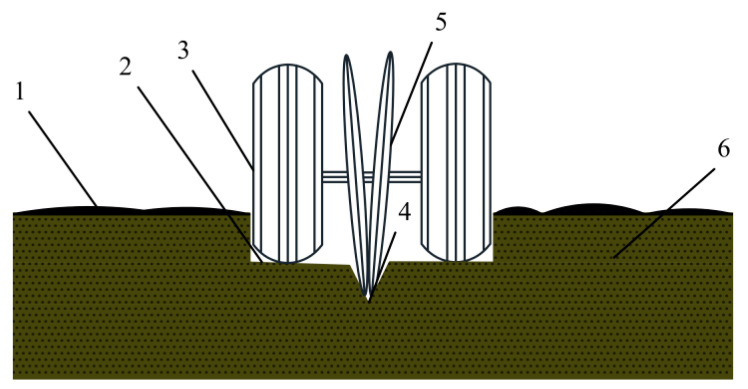
Sowing depth formation schematic. 1. Ground surface. 2. Compaction surface. 3. Depth-limiting wheel. 4. Seed furrow. 5. Disc opener. 6. Soil.

**Figure 2 sensors-24-06331-f002:**
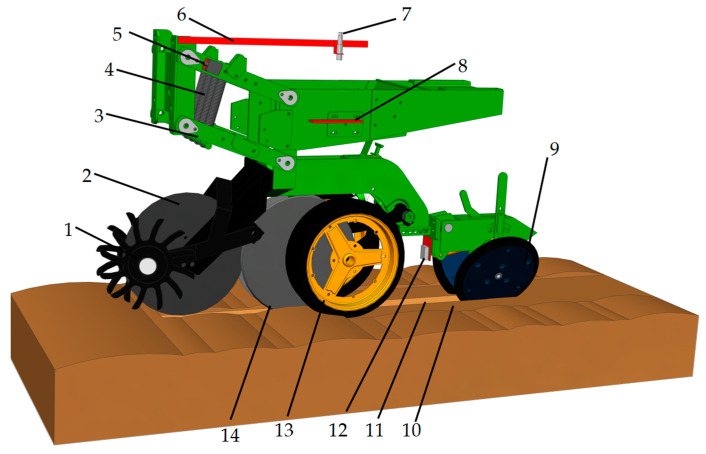
Diagram of sensor layout of sowing depth detection system. 1. Toggle straw wheel. 2. Stubble breaker. 3. Parallel imitation of four links. 4. Pre-tensioning force spring. 5. Angle sensor. 6. Sensor fixing platform. 7. Ultrasonic sensor. 8. Ultrasonic reflective plate. 9. Suppression wheel. 10. Compaction surface. 11. Seed furrow. 12. Laser sensor. 13. Depth-limiting wheel. 14. Disc opener.

**Figure 3 sensors-24-06331-f003:**
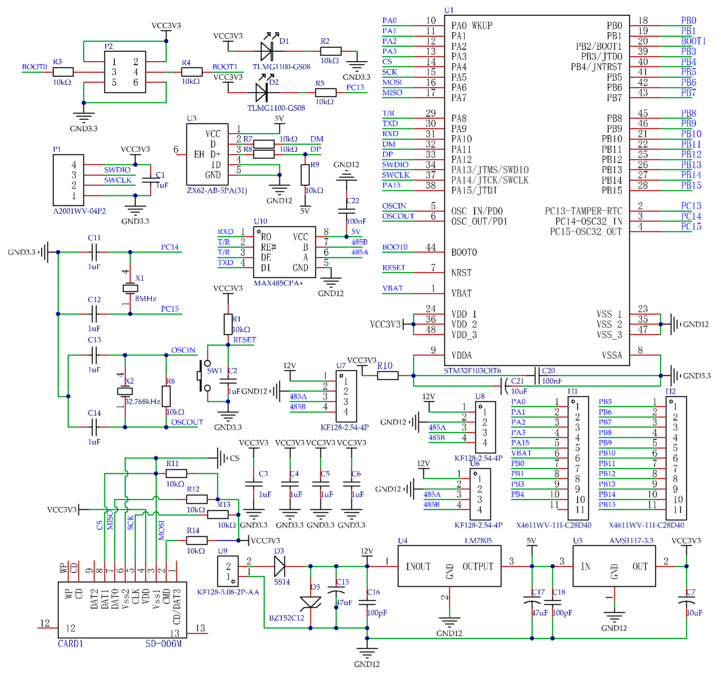
Sowing depth monitoring system circuit.

**Figure 4 sensors-24-06331-f004:**
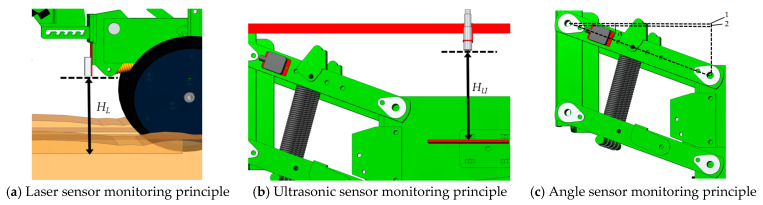
Diagrams of the detection principle of the sensors in the sowing depth detection system. 1. Horizontal plane. 2. Initial height.

**Figure 5 sensors-24-06331-f005:**
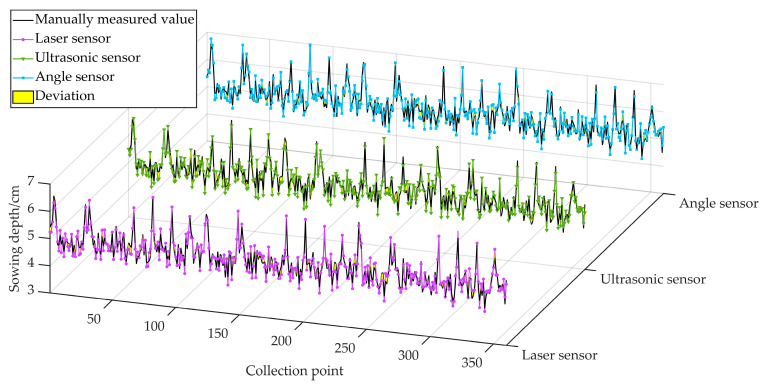
Comparison between the measured values of the three sensors and values measured manually.

**Figure 6 sensors-24-06331-f006:**
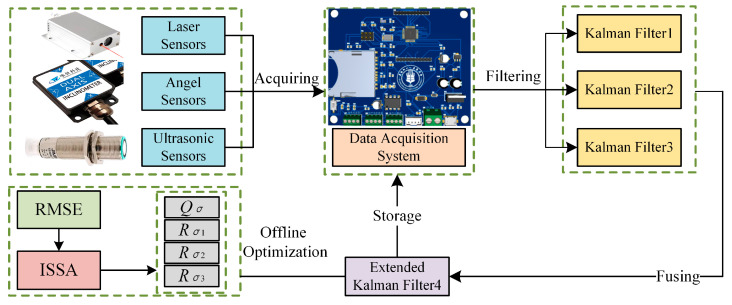
Diagram of principle of multi-sensor data fusion in sowing depth detection system.

**Figure 7 sensors-24-06331-f007:**
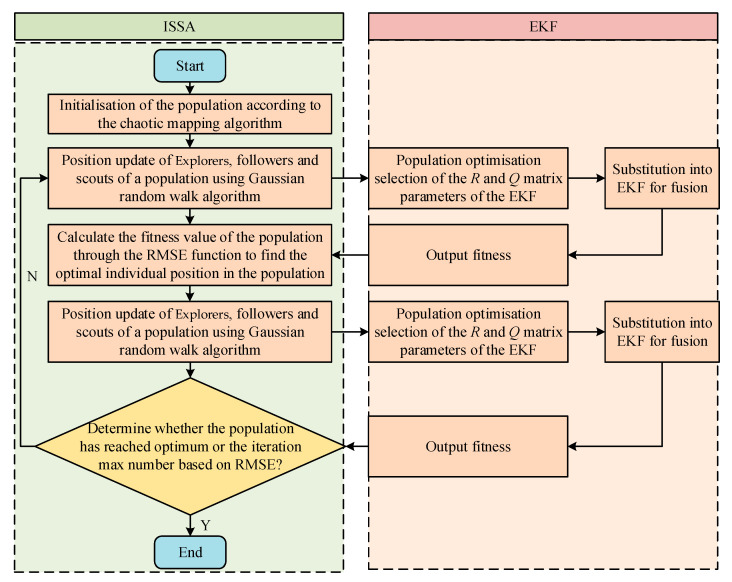
Flowchart of ISSA-EKF.

**Figure 8 sensors-24-06331-f008:**
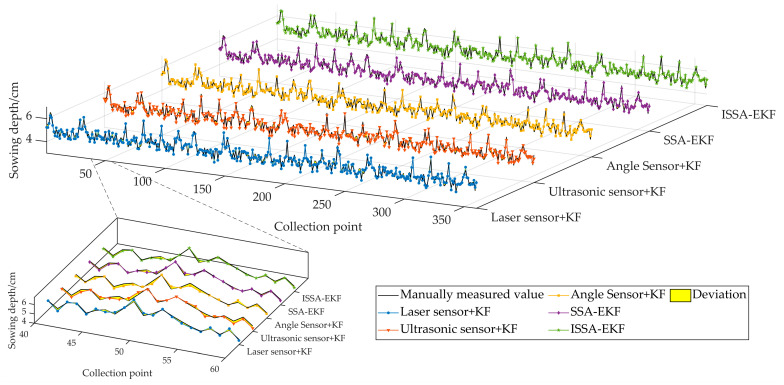
Comparison of 5 filtering algorithms and manual measurements.

**Figure 9 sensors-24-06331-f009:**
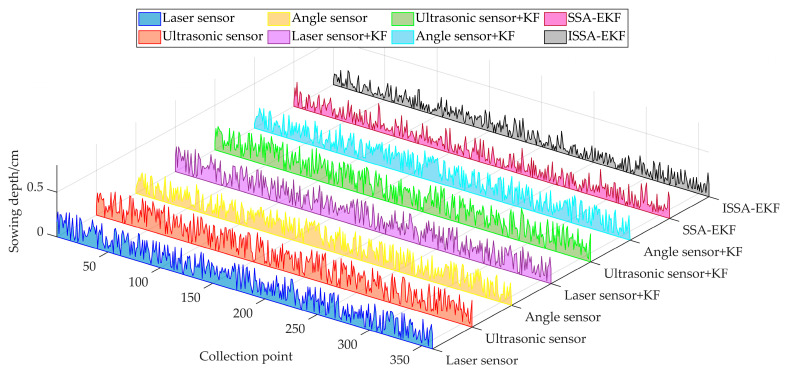
Diagram comparing absolute error of 8 monitoring methods.

**Figure 10 sensors-24-06331-f010:**
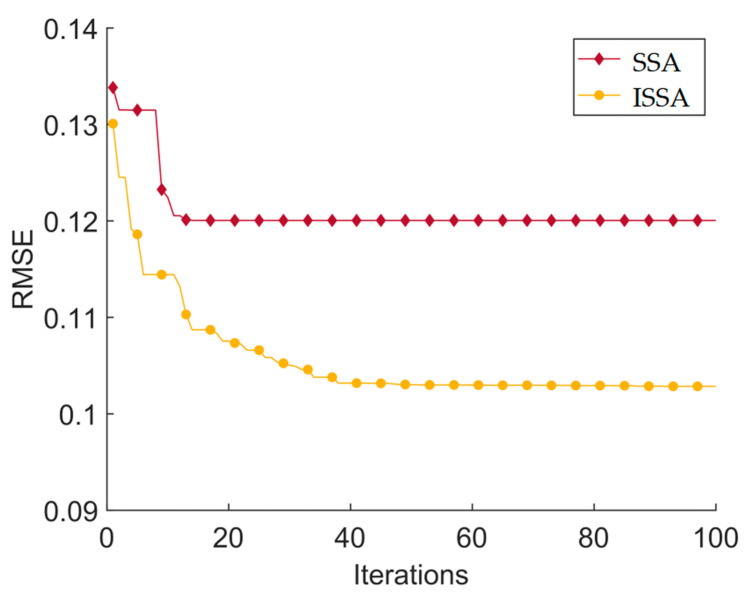
Variation in the fitness values of the two algorithms.

**Figure 11 sensors-24-06331-f011:**
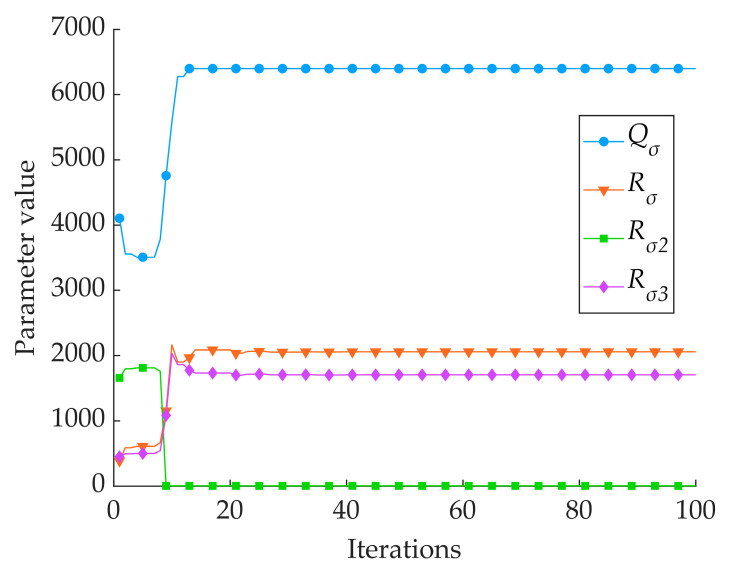
Iterative curve of key parameter optimisation for SSA-EKF.

**Figure 12 sensors-24-06331-f012:**
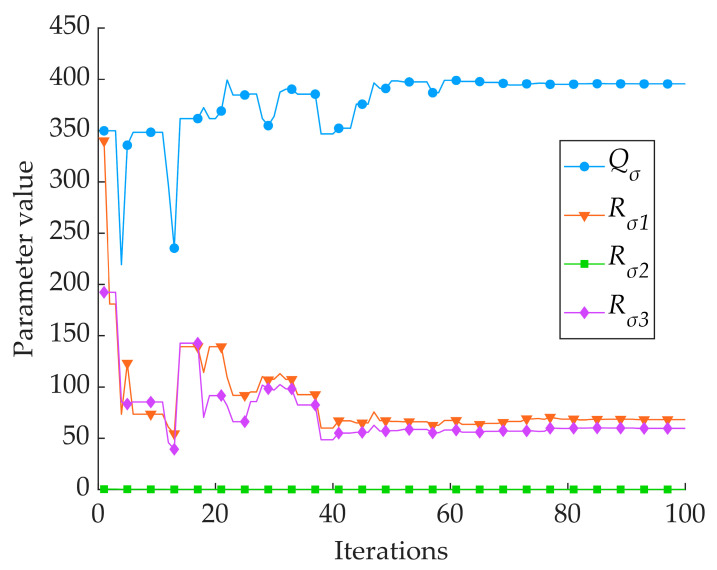
Iterative curve of key parameter optimisation for ISSA-EKF.

**Figure 13 sensors-24-06331-f013:**
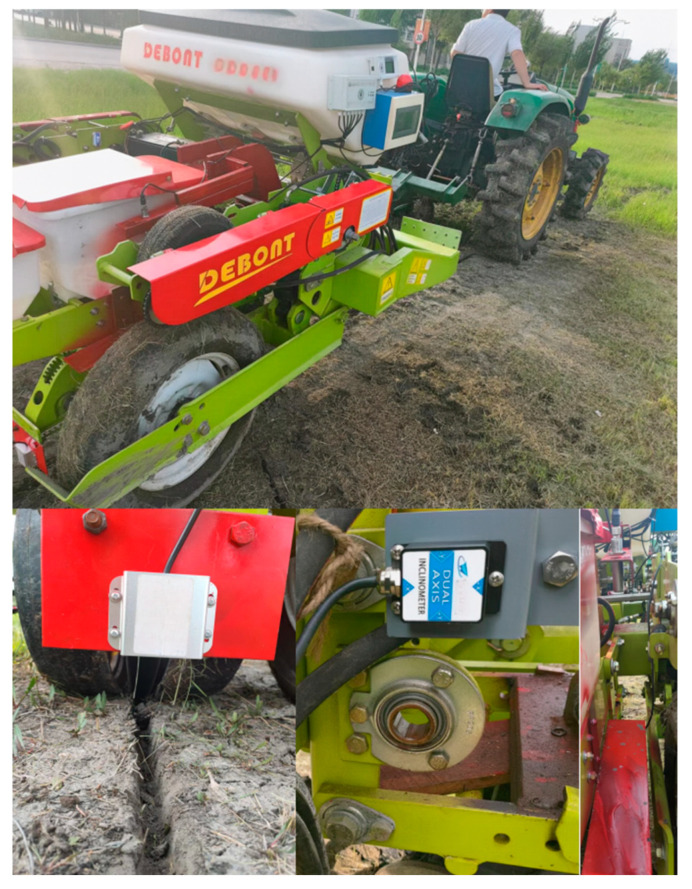
Field test.

**Table 1 sensors-24-06331-t001:** Sensors’ main parameters.

Sensor Type	Main Parameters	Parameter Value
Laser sensor	Measurement accuracy/mm	0.5
	Measurement range/m	0.05~80
	Spot size/mm	6
Ultrasonic sensor	Beam angle/(°)	10
	Sensing range/cm	7~100
	Measurement blindness/cm	0~7
Angle sensor	Measurement resolution/(°)	0.01
	Measurement accuracy/(°)	0.05
	Measuring range/(°)	±90

**Table 2 sensors-24-06331-t002:** Comparison of performance evaluation indicators between the measured values and sensors’ real values.

Data Source	Key Performance Evaluation Parameters		
	*MAE*/cm	*RMSE*/cm	*R*
Laser sensor	0.154	0.179	0.943
Ultrasonic sensor	0.169	0.195	0.930
Angle sensor	0.147	0.169	0.945

**Table 3 sensors-24-06331-t003:** Comparison of results of 5 key indicators of monitoring algorithms.

Data Source	Key Performance Evaluation Parameters		
	*MAE*/cm	*RMSE*/cm	*R*
Laser sensor + KF	0.148	0.172	0.944
Ultrasonic sensor + KF	0.162	0.188	0.932
Angle sensor + KF	0.143	0.166	0.946
SSA-EKF	0.096	0.120	0.971
ISSA-EKF	0.083	0.103	0.979

**Table 4 sensors-24-06331-t004:** Comparison of optimisation results of two optimisation algorithms.

Optimisation Algorithm	Key Parameters				Number of Iterations	Fitness
	*Q_σ_*	*R_σ_* _1_	*R_σ_* _2_	*R_σ_* _3_		
SSA	395	68	60	0.03	20	0.120
ISSA	6400	2057	1705	4	92	0.103

**Table 5 sensors-24-06331-t005:** Field test results.

Theoretical Sowing Depth/cm	Data Source	Key Performance Evaluation Parameters		
		*MAE*/cm	*RMSE*/cm	*R*
3	Laser sensor	0.152	0.175	0.943
	Ultrasonic sensor	0.166	0.189	0.934
	Angle sensor	0.139	0.159	0.953
	ISSA-EKF	0.082	0.103	0.979
5	Laser sensor	0.154	0.178	0.948
	Ultrasonic sensor	0.164	0.188	0.936
	Angle sensor	0.137	0.158	0.955
	ISSA-EKF	0.080	0.097	0.982

## Data Availability

The original contributions presented in the study are included in the article, further inquiries can be directed to the corresponding author.

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
