# Peer review of "Sowing Depth Monitoring System for High-Speed Precision Planters Based on Multi-Sensor Data Fusion"

_sensors, 2024, doi:10.3390/s24196331_

Round 1

Reviewer 1 Report

Comments and Suggestions for Authors

The quality of English needs to be imrpoved signiicantly. Many sentences are long and complex, and it is recommended to break them appropriately to enhance readability.

There is an editing error on line 94.  [Error! Reference source not 94 found.-Error! Reference source not found.]

The article does not clearly give the model of the control chip and sensors used, and the reader is not easy to understand the hardware.

In the circuit of the seeding depth monitoring system given in Figure 3, the circuit arrangement of each module in the schematic diagram is relatively chaotic, and there is no obvious distinction. At the same time, there are many overlapping problems of component symbols and notes in the diagram.

There are many inconsistencies in the line spacing of the article, such as lines 276 to 280. Change is recommended.

RS485 is multi-host communication, three kinds of sensors are using 485 communication, why use three interfaces, rather than using a communication interface.

The use of three sensors for information fusion to obtain tillage depth will significantly increase the cost, compared to the use of one sensor in the end what are the advantages? How much accuracy can be improved?

Comments on the Quality of English Language

Moderate editing of English language required.

Author Response

I would like to express my sincerest gratitude for your thoughtful comments, despite the demands on your time. I am also appreciative of your recognition of our research, which has significantly enhanced the quality of our manuscript. The entire group extends its sincerest gratitude to you. Secondly, the paper has been revised in detail, and a response provided for each comment. The amendments are highlighted in the document for your review. We would like to express our sincerest gratitude once again and wish you continued success in your endeavours.

Comments 1:[The quality of English needs to be imrpoved signiicantly. Many sentences are long and complex, and it is recommended to break them appropriately to enhance readability.]

Response 1:[I am grateful for your feedback. As you have rightly observed, there are numerous issues with the language. In order to address this challenge, we have employed the language services offered by MDPI.]

Comments 2:[There is an editing error on line 94.  [Error! Reference source not 94 found.-Error! Reference source not found.]]

Response 2:[We would like to thank you for your comments. This is one of the format errors that we have identified and subsequently rectified.]

Comments 3:[The article does not clearly give the model of the control chip and sensors used, and the reader is not easy to understand the hardware.]

Response 3:[We are grateful for your input. It should be noted that no particular model of microcontroller unit (MCU) was specified in the original text; however, the MCU model STM32F103C8T6 has been included in Figure 3 for reference. Indeed, the three sensors utilised in this study are illustrated in Figure 6, along with a physical diagram of the system circuit board that houses the microcontroller unit (MCU).]

Comments 4:[In the circuit of the seeding depth monitoring system given in Figure 3, the circuit arrangement of each module in the schematic diagram is relatively chaotic, and there is no obvious distinction. At the same time, there are many overlapping problems of component symbols and notes in the diagram.]

Response 4:[Thank you for your opinion, what you said is very correct, we redrew the circuit diagram, and defined the model and specific value of each component.]

Comments 5:[There are many inconsistencies in the line spacing of the article, such as lines 276 to 280. Change is recommended.]

Response 5:[Thank you for your opinion. The format has been adjusted to solve the same problem in the article.]

Comments 6:[RS485 is multi-host communication, three kinds of sensors are using 485 communication, why use three interfaces, rather than using a communication interface.]

Response 6:[Thank you for your opinion. In order to improve the reliability and anti-interference ability of the system, multiple interfaces are often used to improve the electrical isolation effect. This way prevents a sensor or line failure from affecting communication across the entire system. In addition, in the process of sowing operation, multiple sowing units are usually carried out together. In the future, when expanding the number of sowing units, we will definitely consider your idea. We will use the same interface for the same type of sensor.Therefore, we have added a third paragraph to the discussion section of the article:”This study concerns the selection of RS485 bus communication sensors, which can be used as a communication interface with multiple sensors. In the future, for the expansion of the multi-row sowing monomer sowing depth monitoring system, it may be beneficial to consider an interface access to the same monomer arrangement of sensors. This would allow for the achievement of different sub-systems of individual communication, thus preventing the different sowing monomer information flows from interfering with each other.”]

Comments 7:[The use of three sensors for information fusion to obtain tillage depth will significantly increase the cost, compared to the use of one sensor in the end what are the advantages? How much accuracy can be improved?]

Response 7:[Thank you for your opinion. Please see Table 3 for details of the accuracy improvement. The two data fusion algorithms have greatly improved their performance indicators compared with a single sensor filter. In addition, the multi-sensor can still get a more accurate broadcast depth when a certain sensor is greatly disturbed. Your opinion is also very effective, in the actual development process really needs to consider the cost of the process. After further improving the reliability of the system in the future, we consider removing the ultrasonic sensor or laser sensor from the system and retaining only two of the sensors.Therefore, we have added a third paragraph to the discussion section of the article:”Furthermore, prospective enhancements to the hardware system's reliability and the algorithmic structure will be considered. The objective is to achieve this without compromising monitoring accuracy. One potential avenue for improvement is the removal of the laser and ultrasonic sensors, which are direct measurement sensors that require a fixed frame. This would reduce the system's overall space requirements while simultaneously lowering the hardware costs and system complexity.”]

This is our specific response to your comments. Should any expression be unclear or if there is any ambiguity regarding the original meaning of your comments, we kindly request that you promptly rectify it. We are honoured to engage in communication with you, as it benefits us greatly.

Reviewer 2 Report

Comments and Suggestions for Authors

I have included all comments in the file.

Comments on the Quality of English Language

I have included all comments in the file.

Author Response

I would like to express my sincerest gratitude for your thoughtful comments, despite the demands on your time. I am also appreciative of your recognition of our research, which has significantly enhanced the quality of our manuscript. The entire group extends its sincerest gratitude to you. Secondly, the paper has been revised in detail, and a response provided for each comment. The amendments are highlighted in the document for your review. We would like to express our sincerest gratitude once again and wish you continued success in your endeavours.

Comments 1:[I marked some of the shortcomings, ambiguities or my doubts (such as repeated words, no space between a period and the beginning of the next sentence (or between other objects), no capital letter at the beginning of a new sentence, no reference to an article) with colour directly in the pdf.]

Response 1:[Thank you for your opinion, we looked very carefully and have corrected one by one according to your pdf.]

Comments 2:[The authors use very long sentences. I think that few are ungrammatical, but more important many of them are very difficult to understand. Using short sentences would improve the readability and accessibility of the article.]

Response 2:[Thank you for your comments, there are many problems with the language as you say; we have used the service of MDPI to solve the language problems in the article.]

Comments 3:[Correct subscripts on page 9 (sorry, I can not mark them precisely in pdf).]

Response 3:[Thanks for your comments, this is one of our format errors, which we have corrected.]

Comments 4:[Brackets should be as large as the formula inside them (correct mathematical formulas: (5), (9), (13), (17)).]

Response 4:[Thanks for the comments, the text in the formula has been changed to 10 pit.]

Comments 5:[I can not find citations [9], [16], [17] and [18] in the text.]

Response 5:[Thanks for your comments, I have added the correct index of [9], [16], [17] and [18] in the paper.]

Comments 6:[SSA:The authors use SSA to solve the parameter optimization problem. They cite original paper from 2020, ”A novel swarm intelligence optimization approach: sparrow search algorithm.” by Xue and Shen. However, the algorithm in the peer-reviewed article is a bit different. I tried to implement it for the F 13 function from original paper and found some issues (ξ in (15) from 1 normal distribution?). Is this version of the algorithm taken from some other article (then cite the paper) or created by the Authors themselves (then it should be clearly noted as their achievement)? I would also appreciate this version of SSA in Matlab or Python (if it is written in one of these two programs).

The names of the sparrow groups used in the peer-reviewed paper are different than the ones in the original article. Is this a ploy to emphasize that the version of the algorithm itself is also different or they come from another article? If not, please use original names.

The symbols e and exp are used together. I think it would be better to use one of these two notations.]

Response 6:[Thank you for your comments, first of all, we only used the algorithm logic in the original paper, and did not directly use their code, and secondly, I improved the algorithm, and the specific improvement process has been explained in the article, and it is not all the meaning of the original algorithm. Your second issue is a clerical error on our part, which has been corrected to sparrow population. Your third question is very correct, we add the coefficient of e−βt on the basis of the original formula to dynamically adjust the search step size, so as to balance the global search ability of the algorithm and the local search accuracy, prevent the algorithm from falling into the local optimal prematurely, and ensure that it can steadily converge to the optimal solution in the later stage. Corrected equations (13), (17)]

Comments 7:[Add appropriate spaces in the references.]

Response 7:[Thanks for your comments, we have added spaces in the full references.]

This is our specific response to your comments. Should any expression be unclear or if there is any ambiguity regarding the original meaning of your comments, we kindly request that you promptly rectify it. We are honoured to engage in communication with you, as it benefits us greatly.

Round 2

Reviewer 1 Report

Comments and Suggestions for Authors

Accept